# The Implications of Derisking: The Case of Malta, a Small EU State

**Simon Grima** [1,*] [iD]**, Peter J. Baldacchino** [2]**, Jeremy Mercieca Abela** [2] **and Jonathan V. Spiteri** [1]

[1] Department of Insurance, Faculty of Economics, Management and Accountancy, University of Malta, MSD 2080 Msida, Malta; jonathan.v.spiteri@um.edu.mt

[2] Department of Accountancy, Faculty of Economics, Management and Accountancy, University of Malta, MSD 2080 Msida, Malta; peter.j.baldacchino@um.edu.mt (P.J.B.); jeremy.mercieca-abela.15@um.edu.mt (J.M.A.)

[*] Correspondence: simon.grima@um.edu.mt

**Abstract:** In this study, we explore the emerging derisking phenomenon by identifying and analysing the main factors that are affected by, and the implications of, the derisking process by focusing on the key drivers and implications of derisking specific to Malta. To do this, we carried out 32 interviews with individuals who have a good or excellent level of expertise in derisking and administered a survey, completed by 296 participants who were filtered to ensure their level of expertise, resulting in 285 valid participant surveys. In total, between the interviews and the survey, we had 317 valid participants. Findings showed that to maximise the effectiveness of derisking, one needs to find the right balance of adequately managing risks without extinguishing business needs. This implies a need for the regulations to be balanced and proportionate. This study is a relevant contributor to future derisking to be conducted in Malta and serves as a benchmark for further studies. Moreover, this research project accentuates the need for increased awareness, knowledge and expertise of derisking in Malta. Consequently, the provision of education to professionals is important so that such professionals are able to keep abreast with all the latest developments regarding derisking and AML/CFT (antimoney laundering and combatting the financing of terrorism).

**Keywords:** derisking; drivers and implications of derisking; risk management; antimoney laundering (AML); combatting the financing of terrorism (CFT); financial services; Malta; small EU state; proportionality

## 1. Introduction

The implementation of a derisking strategy is a recent trend; many banks and financial institutions worldwide are undertaking derisking to simplify their business models, minimise their risk exposures and to comply with the increasing AML/CFT (antimoney laundering and combatting the financing of terrorism) regulations. Other key driving factors, which have led to many banks around the world opting for derisking, include a steady increase in costs necessary to ensure compliance with AML/CFT regulations, rising fines and penalties imposed for noncompliance and the lower profitability of certain customer bases.

Derisking is a process which involves the closure of the bank accounts of, and/or the termination of relationships with, clients categorised as being high risk after considering the risk appetite of the organisation (Durner and Shetret 2015). Nisar (2016) provides some examples of these types of clients, which can include money service businesses (MSBs), correspondent banks, embassies, international charities, nonprofit organisations (NPOs) and multinational corporations (MNCs). Derisking can be carried out in two ways: (1) Wholesale derisking involves the derisking of an entire category of

customers (known as a customer base) or a particular sector. Examples of sectors which are often subject to wholesale derisking are MSBs, NPOs, charities and the defence sector (Durner and Shetret 2015). Alternatively, (2) derisking on a case-by-case basis is when a bank first considers the attributes of an individual customer and the relationship the bank has with such a customer, and then carries out a customer risk assessment. Based on all the information obtained, the bank finally decides on whether or not to carry out derisking (Artingstall et al. 2016).

Moreover, as part of their derisking strategy, banks and other financial institutions may also decide to limit the range of financial services they are willing to provide to these types of clients (Haley 2017). However, although it is thought that by engaging in derisking and terminating relationships with the so-classified high-risk clients, banks and other financial institutions are able to minimise their risk exposures, this has significant negative repercussions on clients and such clients will end up bankless, ultimately leading to increased financial exclusion (Durner and Shetret 2015).

Haley (2017) mentions three main forms of derisking:

1. When banks and other financial institutions decide to close the bank accounts of certain individuals or firms or when banks and other financial institutions opt not to open bank accounts for these individuals or firms in the first place. This form of derisking is also known as debanking. The costs arising from this type of derisking are largely borne by the individuals. Furthermore, these costs are usually only limited to restricting financial integration and do not represent a systemic risk. This holds true assuming that domestic banks are able to provide services for transactions that are not cross-border in nature.

2. When banks and other financial institutions restrict or completely withdraw services from money service businesses (MSBs; Haley 2018). MSBs are nonbank institutions that offer a variety of financial services at a much lower cost when compared to traditional banking institutions (Durner and Shetret 2015). Some of the services that MSBs provide include the transmission of money, currency exchange and the checking of cheques (Durner and Shetret 2015). One of the most noteworthy types of MSBs is money transfer organisations (MTOs). These are defined as financial companies, typically not banks, that are involved in the crossborder transfer of funds (IMF 2008, 2012). When it comes to the transfer of funds, MTOs can either use their own internal systems or access "crossborder banking networks" (IMF 2008, p. 9). MTOs provide services which are vital when it comes to the efficient transferring of remittances worldwide. Therefore, if MTOs are derisked, this implies that other informal channels for the transferring of funds will emerge. These informal channels may open pathways for money laundering and financing of terrorism, thus posing a significant risk to efforts being made regarding AML/CFT (antimoney laundering/combatting the financing of terrorism; Haley 2017).

3. The loss of correspondent banking relationships (CBRs; Haley 2018). These are an arrangement under which one bank (correspondent) holds deposits owned by other banks (respondents) and provides payment and other services to those respondent banks (CPMI 2003, p. 16). Essentially, in a CBR, domestic banks hold accounts in large international banks, and, in turn, such large international banks will provide access to a global payment and settlement system. As a result, if CBRs are severed, domestic banks will be unable to process payments and provide services to their clients. This represents a potential systemic risk and could lead to an inefficient global payment system (Haley 2017).

## 2. Rationale for the Study

Over the past few years, the derisking practice has also gained popularity in Malta, and, in fact, there are quite a few domestic and international banks operating in Malta that have engaged in, or are in the process of engaging in, derisking.

Given that at present, very limited research regarding the implications of derisking vis-à-vis the Maltese scenario has been carried out, this implies the need for such a study. This study fills such a

gap by identifying and subsequently laying out the analysis of the factors that are affected by, and more importantly, the drivers and implications of, the derisking process.

Although we herein highlight the main drivers of the derisking process in a small EU state and island, as suggested by various researchers such as Bezzina et al. (2014), Briguglio (1995), and King (1993), given the jurisdiction's size and nature, the results can be used as small-scale laboratory test findings for more complex politics, regulations and policies of larger countries. Therefore, the findings of this study are of significance to international policymakers, regulators and governments.

## 3. The Objective and Research Questions of the Study

The researcher has one primary objective: to identify and analyse the implications that are affected by the derisking process. Moreover, underpinning this objective are two research questions:

RQ1: What are the main implication variables of a derisking strategy within the context of Malta?

RQ2: What are the key effects of derisking with regards to the Maltese scenario?

In doing this, we also check whether the drivers differ with the different demographics.

The aim is to understand what the implications variables are, which should be looked at when carrying out a derisking strategy, that is, a checklist to help risk managers develop a risk matrix and measure the implications of derisking as one single qualitative or quantitative value. This can also serve as a tool for policymakers.

## 4. Why Was Malta Chosen as the Focal Point of This Study?

As noted above, over the years, many leading and well-known researchers such as King (1993), Briguglio (1995), Baldacchino (2006), Bezzina et al. (2014), and Xuereb et al. (2019) have made use of small island states like Malta in order to understand the complex financial environments, dynamics, politics, policies and regulations of larger countries (Xuereb et al. 2019). Therefore, small island states are utilised as laboratories on a smaller scale (Xuereb et al. 2019; Pirotta 2001).

Moreover, the principle of proportionality is of utmost importance to ensure that smaller states still have a voice. Proportionality means that the EU should only take the action that is necessary to achieve its aims. This principle is set out in Treaty 5 of the EU under Article 5. In reality, when it comes to the EU's decision-making process, it is usually the larger states that have more prominence and are able to influence the outcome of final decisions. Therefore, in practice, the concept of proportionality is not always adhered to and is not effectively managed (Xuereb et al. 2019). Adopting an approach which is proportional involves taking into account the size, nature, systemic importance, complexity and risk profile of the stakeholders involved and the services being offered. Therefore, it is important that regulations are tailored according to such requirements. This leads to lower costs related to compliance and a decreased regulatory burden for smaller and less complex organisations that might otherwise experience an undue reduction in their competitiveness. Consequently, this study is of significant importance as it gives smaller states the opportunity to have a much-needed voice and allows further understanding to be obtained regarding the implications of derisking vis-à-vis smaller jurisdictions (Xuereb et al. 2019; Lautenschläger 2017; Grima et al. 2016).

## 5. Literature

### 5.1. The International De-Risking Landscape

#### 5.1.1. Traditional Risk Management

The traditional risk management approach used reactive models, defined as a managerial process, an administrative process or a decision-making process of planning, organizing, leading, and controlling the organization's activities to lower the adverse effects of any accidental losses that a firm might incur. It was more of a reporting and monitoring function.

It is a risk-based approach, which places responsibility on various departments and business units and the focus is on pure risk (hazard risk where the consequences may or may not be losses) and refers to individual risks as though they do not interact with each other. Therefore, this approach does not align with the risk management needs of an organisation since risks should be treated as a whole (taking into consideration the direct and indirect effects). The result is, therefore, unsatisfactory due to different types of risks being managed and treated independently. This fragmented approach to risk cannot work within an aggregated approach to risk throughout an organisation (Head 1973).

The complexity, volatility and unpredictability of the current economic and social environment are a constant reminder of the many risks we need to face to reach objectives. The traditional approach is, therefore, no longer sufficient, and it is necessary for today's companies to invest significant resources to identify, measure and manage risks and ensure longevity. The risk management approach has a major role in an organisation's ability to avoid, reduce and turn risks into opportunities.

### 5.1.2. The Context of Derisking

Over the past years, regulatory scrutiny in the global financial services sector has significantly increased, especially regarding financial crime and AML/CFT regulation (Badour et al. 2017). The reaction of banks and other financial institutions to this new regulatory environment was to strengthen and invest more in their compliance and risk management functions (Badour et al. 2017). As costs associated with regulation and compliance continued to increase, some banks and other financial institutions started carrying out cost–benefit analyses with regards to certain activities and operations they carry out. Such banks and other financial institutions found that in some cases, it is more beneficial to stop servicing certain customer segments and/or to exit specific product lines (Badour et al. 2017).

In the current global landscape which comprises different actors concerned with derisking, there are many of such actors (e.g., the G20) that view this phenomenon as being an issue that is related to financial stability (HSC and ECNL 2018). This is because derisking has a considerable impact on CBRs, which may, consequently, negatively impact trade financing and economic development. Other actors such as the World Bank believe that derisking is broader in scope and may even affect financial integrity and inclusion (HSC and ECNL 2018). It is important that banks and other financial institutions do not only aim to safeguard their margins but also act as "good global citizens [by] providing a public service" (HSC and ECNL 2018, p. 12).

On the other hand, according to regulatory framework architects, derisking is not the result of stricter AML/CFT regulations, but principally, the result of banks and other financial institutions not properly applying and/or misinterpreting the requirements related to AML/CFT. At the same time, such banks and financial institutions complain that clean money is disappearing into banking channels that are unregulated (HSC and ECNL 2018).

### 5.1.3. The Political Landscape

Long-established political concerns with regards to financial crime and terrorism collide with an increased demand for regulation that creates "legal, political and operational imperatives that result in derisking" (HSC and ECNL 2018, p. 13).

The pressure of ever-increasing international policy responses when it comes to financial crime and terrorism has transformed how the global financial system works and operates (HSC and ECNL 2018). In the 1980s, international architecture regarding the surveillance and control of customers was introduced. Following this was the establishment of enforcement mechanisms, which mainly comprise of economic and financial sanctions. As a result, financial services providers are now also being held responsible for monitoring activities that are carried out by their customers (HSC and ECNL 2018).

The Financial Action Task Force (FATF) is one of the most important and powerful international organisations with respect to derisking. The FATF was established in 1989 by the G7 countries with the primary aim of developing a framework to combat money laundering that could be applied on an

international level. Upon its inception, this organisation imposed obligations on three levels (HSC and ECNL 2018):

1.  Firstly, financial services providers had to start conducting due diligence and identifying customers engaging in illicit activities.
2.  On a domestic level, states had to start investigating suspicious transactions and financial crime.
3.  Finally, on an international level, cross-border cooperation with respect to financial crime investigations and the freezing of assets of individuals involved in such financial crimes were encouraged.

Following the September 11 attacks in 2001, the mandate of the FATF was broadened in scope to include the combatting of the financing of terrorism so as to stop money from reaching terrorist organisations. Here, the ultimate aim is that of increasing peace and security on a global scale (HSC and ECNL 2018). The FATF also has the power to ensure that countries implement their recommendations. Countries that do not comply are blacklisted from the global financial system until they achieve full compliance once again (HSC and ECNL 2018).

The logic behind the transformation that the FATF underwent over the years can also be applied to the AML/CFT framework, which focuses on areas of failure and does not take into account other concerns such as human rights and accountability (HSC and ECNL 2018). The unintended spill-offs of this approach, such as derisking, are oftentimes tolerated and not questioned. It is only when derisking leads to funds being pushed into shadow banking channels that it starts becoming a concern for many prominent stakeholders (HSC and ECNL 2018).

5.1.4. The Regulatory Landscape

To minimise the risk of noncompliance, the banking and financial services sectors have opted to internalise AML/CFT regulations and are regulated both statutorily and on a "self-regulation" basis. Individually, banks have also developed numerous internal mechanisms to ensure compliance and mitigate risk (HSC and ECNL 2018).

The role of central banks is to make sure that AML/CFT regulations that have been incorporated into the banking regulatory framework are actually applied in practice (HSC and ECNL 2018). "Breaches of sanctions regimes or failures to conduct proper due diligence" (HSC and ECNL 2018, p. 16) can lead to the imposition of fines and even the threat of criminal prosecution. In this regard, the biggest driver of derisking is not the cost of financial penalties, but the enforcement actions' reputational costs if the banks involved are allegedly found to be colluding with terrorist groups or organised criminals (HSC and ECNL 2018). Consequently, banks are revising their risk appetites and implementing more widespread derisking (HSC and ECNL 2018).

The internal compliance mechanisms that banks are introducing to avoid the risks and fines for not being compliant are usually developed by individual third-party financial services providers (HSC and ECNL 2018). Therefore, determining whether individual clients or entire business segments are perceived as being high-risk is a commercial decision based on the algorithmic decision-making of a third-party actor and not on an impartial and objective risk assessment (HSC and ECNL 2018). The industry that provides such compliance services has thus normalised the derisking process (HSC and ECNL 2018).

When it comes to derisking, it is important that relevant stakeholders take into consideration not only certain risk management tools that are provided by the industry (HSC and ECNL 2018), but also the "political economy of the compliance sector" (HSC and ECNL 2018, p. 16).

Derisking is not always a rational process based on sound cost–benefit analyses. It can be "the result of (highly politicised) processes predicated on the creation of 'suspect communities' to maintain an adequate supply of risk to be managed" (HSC and ECNL 2018, p. 17). This is not an appropriate baseline for the prevention of financial crime and terrorism and leads to statutory regulation not being applied on a proportionate basis (HSC and ECNL 2018).

### 5.1.5. The Relationship between Money Laundering, Corruption and Bribery

Money laundering and corruption are two serious offences which are very much linked to one another. The prevalence of money laundering in any particular country is a clear indicator that corruption is also prevalent and vice versa (Mugarura 2016). The ideal environment in which money laundering and corruption thrive is one which is characterised by a lack of adequate oversight by regulatory authorities, bad governance and weak laws and regulations (Mugarura 2016).

One very common type of corruption is bribery. The underlying objective of bribery is to influence the opinions and/or behaviour of certain individuals such as public officials. Bribes often entail gifts and rewards; however, they are not always monetary in value. One example of a nonmonetary bribe is hospitality. The prevalence of bribery within a particular country leads to both short-term and long-term economic and reputational damage (Clark 2019; Consiglio and Grima 2012).

Since money laundering and corruption are intertwined, it is important that countries not only introduce anticorruption policies but also adopt robust AML strategies and make sure that such strategies are effectively implemented and enforced. Proper enforcement of AML strategies helps to hold corrupt public officials accountable for their actions (Mugarura 2016).

The manifestation of money and corruption hinders the ability of individual countries to pursue development goals on a national level and also facilitates crimes such as human, drug and small arms trafficking, terrorism financing, as well as prostitution (Mugarura 2016).

Corruption and bribery are often mistaken for one another or used interchangeably since their meanings are very similar. However, to prevent, manage and mitigate the problems they cause, it is important to understand the differences between them.

Corruption is defined as any actions by a person in a high position to take an unfair advantage, usually for personal gain. This is the abuse by persons with entrusted power for private gain, which can be of any size, commercial or political.

Bribery, on the other hand, is a type of corruption. An example is offering gifts to individuals so as to influence their opinions or behaviour (Consiglio and Grima 2012).

### 5.1.6. A Sustainable Model for AML in the United Nations Development Goals

In 2015, all the member states of the United Nations (UN) adopted 17 sustainable development goals (SDGs) that focus on tackling a number of prominent issues faced by all countries around the world. Since money laundering destabilises domestic economies, it needs to be scaled down so that such SDGs can then be achieved (Dobrowolski and Sulkowski 2019).

According to Dobrowolski and Sulkowski (2019), money laundering "distorts the efficient and effective outcomes of economic activities and serves as a catalyst for other types of crime, including corruption, fraud, drug trafficking, and terrorism" (p. 1). Therefore, money laundering can have a crippling effect on economies worldwide and leads to distorted international finances. Furthermore, money laundering significantly decreases integrity, in turn leading to reduced transparency and accountability. Ultimately, this results in higher levels of public mistrust, adversely impacting global markets and stifling innovation and development (Dobrowolski and Sulkowski 2019).

Supreme audit institutions (SAIs) have the power and capacity to audit, investigate and evaluate AML systems and antimoney laundering organisations (AMLOs) to assess whether such systems and organisations are effectively fulfilling their obligations and tasks. In addition, SAIs have the responsibility of providing decision-makers and governments with accurate and reliable information, which allows them to establish AML systems which are efficient and effective. Therefore, SAIs promote good governance. Notwithstanding, the capacity of SAIs is sometimes limited due to a lack of standards and guidelines regarding the auditing of AML systems and organisations (Dobrowolski and Sulkowski 2019).

To combat money laundering in an effective manner, Dobrowolski and Sulkowski (2019) proposed the use of the MAMA (multiple antimoney laundering activity) model as a universal framework of AML and an assessment tool for auditors. This model is beneficial because it can assist all types of

organisations from all countries around the world. Moreover, this model allows different organisations to compare their different AML systems, meaning that benchmarks can be established (Dobrowolski and Sulkowski 2019). The MAMA model consists of 8 criteria (Dobrowolski and Sulkowski 2019):

- Five of these criteria are "enablers" that focus on the core functions of AMLOs—(1) leadership, (2) people, (3) policy and strategy, (4) partnership and resources, and (5) processes.
- The other three criteria are "results" that focus on the achievements of AMLOs—(1) employee results, (2) institutional stakeholders results, and (3) society results.

## 5.2. The Drivers of Derisking

When analysing the driving factors of the derisking process, it is imperative to take into account recent developments in the spheres of banking and regulation (Artingstall et al. 2016). The approach being adopted regarding risk is changing as the priority for banks and other financial institutions is to decrease their costs. Furthermore, banks and financial institutions aim to reduce risk-weighted assets and slim down their business in order to lower their risk profiles (Artingstall et al. 2016).

The derisking process is driven by an amalgamation of factors such as the ever-increasing AML/CFT regulations, higher compliance costs, more stringent and demanding prudential requirements, increased risk management controls and low profitability of certain business areas (Langthaler and Niño 2017).

To facilitate comprehension, these aforementioned drivers of derisking and other factors are going to be grouped under specific subheadings and explained in more detail in the following pages.

## 5.3. Decreasing Risk Appetites

In response to the 2008 global financial crisis, many banks started scaling back their risk appetites and adopting a risk-based approach (RBA) to AML/CFT. By adopting an RBA to AML/CFT, banks reduced the extent to which the services they provide to customers can be abused for the purpose of financial crime. This is done by such banks discontinuing relationships that they believe present too high a risk of this abuse (Artingstall et al. 2016). The reduction in the risk appetites of banks and other financial institutions has led to more widespread derisking of clients that do not fall within such decreased risk appetites.

The exiting of relationships with customers perceived as presenting the highest risk helps minimise counterparty risk. Consequently, banks can improve their risk profile and ensure compliance with the RBA to AML/CFT regulations, which many regulators advocate in favour of (Badour et al. 2017). Furthermore, it is beneficial for banks to have strong practices associated with risk management in place as this leads to reduced regulatory risk and a lower likelihood of regulatory action (Badour et al. 2017).

It is paramount that any organisation determines its level of risk appetite as this ensures that effective decisions are taken when determining strategies to manage risk by, for example, transferring or accepting part of or the whole risk and determining which tools to utilise when doing this. ISO 31000 defines risk management as being "the amount and type of risk that an organisation is willing to pursue or retain" (ISO 2009). Such risk is pursued and retained so that the organisation's strategic objectives can be achieved (Alix et al. 2015; Bromiley et al. 2015; Lamanda and Voneki 2015; as cited in Zhang 2016). In the context of derisking, banks and other financial institutions usually derisk the clients that are deemed to be too high-risk (in the light of the appetite of the shareholders, transmitted through the board of directors) and fall outside of their risk appetite. Moreover, the implementation of proper risk mitigation strategies decreases the adverse effects of risks that an organisation is exposed to.

Risk appetite is sometimes also defined as that amount of financial crime residual risk that a bank or financial institution is willing to have onboard and the view of how efficiently and effectively it can manage such risks (Artingstall et al. 2016). Risk management involves two complementary types of mitigation—enhanced due diligence and continuous monitoring. Beyond such mitigation strategies, interventions will then involve some form of derisking (Artingstall et al. 2016).

RBAs with respect to financial crime identify and analyse the risk that is associated with factors that include the type of business involved, the sector, the occupation, jurisdiction and political

risk, distribution channels and the products or services required by the customer (Artingstall et al. 2016). These customer risk ratings allow banks to assign scores to specific customers and then determine whether the financial crime risks involved are acceptable and fall within the particular bank's predetermined parameters. This all depends on the financial crime risk appetite that the bank has (Artingstall et al. 2016).

### 5.4. Increasing AML/CFT and Regulatory Compliance

#### 5.4.1. AML/CFT Compliance

With AML/CFT regulations always increasing and becoming ever more complex, this has resulted in higher costs related to compliance (Haley 2017). A survey on global antimoney laundering carried out by KPMG in 2014 found that 78% of respondents, representing the most prominent global banks, stated that their banks reported a substantial increase in AML compliance, with 22% of such respondents revealing that their banks experienced a 50% increase from the years 2011 to 2014 (KPMG 2014). In 2014, HSBC spent $800 million on risk management and compliance, which was about a $200 million increase from the previous year (Arnold and Hughes 2014).

The main factor which has led to rising compliance costs is the imposition of enhanced customer due diligence (CDD) requirements (Haley 2018). These require banks to follow Know Your Customer (KYC) guidelines, which mandate that banks need to identify and verify their customers and also the ultimate beneficial owners of their legal entity customers (Langthaler and Niño 2017). Additionally, banks must comprehend the nature and purpose of the relationships with their customers and conduct monitoring that is ongoing. In the case of clients and countries that are both perceived as being higher risk, KYC requirements and due diligence can be very costly and time-consuming (Langthaler and Niño 2017).

Rising AML/CFT compliance costs have a magnified effect on jurisdictions that have smaller and more restricted banking markets as such jurisdictions are not always able to spread the fixed cost component of these AML/CFT compliance costs over a large enough number of transactions (Haley 2017).

In addition, rising compliance costs may disincentive larger banks to uphold certain interbank relationships that provide additional cover or more transactional options. This negatively affects small and medium-sized banks, which may end up having to stop servicing some of their customers and other banks (Artingstall et al. 2016).

In their study entitled Drivers and Impacts of Derisking, Artingstall et al. (2016) analysed banks that have engaged in derisking and closed several of their customers' bank accounts. The banks investigated were not willing to share a portion of these increased compliance costs for bank accounts with their customers by using the customers' risk rating as a basis. If these compliance costs could be calculated for each individual customer, there might have been customers willing to pay a premium to maintain their bank accounts (Artingstall et al. 2016). This is essentially a form of market failure, where if costs and resources are allocated more efficiently against the actual money laundering and terrorism financing risks involved, banks would need to derisk less (Artingstall et al. 2016).

Badour et al. (2017) argue that the derisking process actually leads to lower compliance costs and allows banks to operate with a greater level of efficiency. Exiting relationships with high-risk customers that typically require considerable compliance resources allow such resources to be used elsewhere. By focusing on fewer product lines that do not involve high compliance costs, this will allow banks to operate more effectively and simultaneously, also possibly boosting their profitability (Badour et al. 2017).

#### 5.4.2. Regulatory Compliance

According to Durner and Shetret (2015), regulatory approaches often diverge across "state, national and international jurisdictions" (p. 10). Such divergence is usually considered to be an

invisible cost associated with globalisation and results in increased costs related to compliance (Durner and Shetret 2015).

Apart from the significant fines, agreed settlements of regulatory action may include the imposition of certain business restrictions as part of such regulatory settlement for the organisation under consideration. This is known as regulatory derisking (Artingstall et al. 2016).

An example of a regulatory settlement reached when certain AML/CFT requirements are breached is a deferred prosecution agreement (DPA; Artingstall et al. 2016). With regards to DPAs, banks agree to voluntarily abide by several conditions in exchange for the suspension of their criminal charges. The conditions, stipulated in a DPA, comprise of financial penalties and improvements that must be made with regards to particular compliance measures and cooperation with the applicable regulatory and law enforcement agencies (Durner and Shetret 2015).

Costs associated with compliance of a regulatory nature may ultimately end up being shifted to customers in the form of increased fees, reduced availability of services and restricted credit (Durner and Shetret 2015). Low-income individuals and businesses that have low profit margins may not be able to sustain these newly added costs, leading to services being discontinued and these individuals and businesses being financially excluded (Durner and Shetret 2015).

### 5.4.3. Rising Fines and Penalties

The imposition of significant fines and penalties on banks and financial institutions for noncompliance with and breaches of AML/CFT requirements is another main driver of derisking (Haley 2018). Fines and penalties may also be issued if there are deficiencies in AML/CFT compliance and if sanctions are violated (Durner and Shetret 2015). Table 1, depicted below, taken from Haley (2018, p. 5), shows some examples of fines and penalties that were imposed on international banks for noncompliance.

After the end of the 2008 global financial crisis, regulators worldwide started facing increasing pressures to hold organisations accountable for misconduct (English and Hammond 2014). Nevertheless, such regulators have stipulated that the imposition of substantial fines and enforcement actions is only for the most egregious and sustained offenders (Durner and Shetret 2015).

**Table 1.** Fines and penalties imposed for noncompliance.

| Bank | US$ (Billions) | Date |
|---|---|---|
| HSBC | 1.9 | December 2012 |
| J.P. Morgan Chase | 1.7 | January 2014 |
| BNP Paribas | 8.9 | July 2014 |
| Commerzbank | 1.5 | March 2015 |

Source: author's compilation.

### 5.4.4. Declining Customer Profitability

Low profitability of certain customer bases is another factor which drives derisking. Indeed, there are some cases where "it is simply not worth the hassle and costs" (Durner and Shetret 2015, p. 17) to engage with a particular customer base. If the profitability associated with a specific customer base does not compensate for the higher risk, the additional costs that must be incurred and the further AML/CFT controls that must be implemented, the customer base is derisked (Durner and Shetret 2015).

Customer profitability is also important when it comes to assessing CBRs. Nowadays, banks and other financial institutions engage in a practice known as Know your Customer's Customer (KYCC), which means that the responsibility of banks and other financial institutions is not only limited to their customers but also the customers of their customers (Durner and Shetret 2015).

Since KYCC represents an additional risk for banks and other financial institutions engaging in CBRs (especially in high-risk jurisdictions) and is considered as being a costly and rigorous process, such banks and other financial institutions often resort to derisking (Durner and Shetret 2015).

Given the fact that KYCC is a highly intensive and expensive process, banks and other financial institutions may decide that the risk involved is not worth the reward with regards to certain high-risk customer bases. If this is the case, the subsequent closure of the bank accounts of these clients represents a clear market failure, especially considering the financial inclusion benefits derived if these sectors continue being serviced (Durner and Shetret 2015).

### 5.4.5. Reputational Concerns

The consequences of noncompliance are not only limited to fines and penalties. Other types of enforcement actions, e.g., limitations, when it comes to the provision of certain services can also result in the organisation incurring high financial costs (English and Hammond 2014).

Certain enforcement actions can lead to a bank incurring reputational damage, and this can have devastating effects. As regulatory scrutiny continues to increase, so do the chances of banks being found guilty of noncompliance with AML/CFT procedures and sanctions. Often, there is an element of uncertainty and concern regarding the bank's ability to survive the enforcement action. This can have an adverse effect on the relationship that such a bank has with investors, meaning that the share price may be impacted (Durner and Shetret 2015).

Since banks provide crucial services to vulnerable communities, this implies that the derisking of such communities can have negative ramifications when it comes to public relations (Durner and Shetret 2015). Therefore, the continued servicing of these vulnerable communities can lead to banks obtaining "reputation returns" (Durner and Shetret 2015, p. 12).

### 5.4.6. Higher Capital and Liquidity Requirements

A key driving factor that also influences the derisking carried out by banks and other financial institutions is the imposition of increased capital requirements and liquidity thresholds following the 2008 global financial crisis (Haley 2017).

Such higher requirements and thresholds have created an environment in which it is harder for banks to maintain profitable customer relationships, with some banks resorting to deleveraging (Artingstall et al. 2016). Moreover, many banks worldwide have opted to undertake a strategic review of their main business and functions, which has led to such banks focusing more on businesses that are the most core (FCA 2016).

### *5.5. The Factors and Implications of Derisking*

The main factors, implications, and effects of derisking, identified through various sources of the existing literature, are described below:

### 5.5.1. Shift of AML/CFT Risk and Shadow Banking Channels

When banks and other financial institutions engage in derisking, this results in a shift of AML/CFT risk. This is because if CBRs are terminated, affected clients will then have to start relying on smaller banks and credit institutions to obtain the financial services they require. The problem with this is that such smaller banks and credit institutions may not have the expertise and capacity necessary to service higher risk clients (Durner and Shetret 2015).

Some customers may have no other choice but to turn to shadow banking channels to address their financing needs (Durner and Shetret 2015). What makes these alternative channels an attractive proposition is their ease of access and low costs (Langthaler and Niño 2017). However, this does not always hold true—if these underground channels are the only way certain customers can obtain funds, this implies that such customers may have to incur higher costs to acquire finance (Durner and Shetret 2015). Moreover, these channels present a higher risk as they are not very regulated and monitored, suggesting that there is less transparency and accountability towards customers (Durner and Shetret 2015).

These aforementioned points create a regulatory paradox. This is because the introduction of stricter regulation and increased AML/CFT compliance requirements has resulted in the implementation of more derisking strategies by banks and other financial institutions, and operations are being pushed to markets that are not properly regulated and monitored (Langthaler and Niño 2017).

### 5.5.2. Financial and Socioeconomic Implications

The worst effects of the global phenomenon of derisking are felt the most by smaller countries that typically have low volumes of crossborder transactions. Countries that are also at a high level of risk are those that have a significant dependence on remittance payments (Haley 2018).

The derisking process can lead to reduced international trade finance. At times, international banks may decide to terminate certain CBRs with smaller foreign banks and close their correspondent bank accounts. This subsequently limits the access of such foreign banks to foreign currencies. Access to foreign currencies is vital because it is necessary for international trade to be conducted and it also facilitates international investment (Durner and Shetret 2015). Therefore, if a country's access to such currencies is lowered, this leads to the whole country being debanked, adversely impacting economic growth (Durner and Shetret 2015).

According to Langthaler and Niño (2017), tourism is one of the industries that suffers the most when a country's access to foreign currencies is restricted, especially in the case of Caribbean countries. If foreign exchange services become more difficult to access, tourism flows and related investments will decrease (Langthaler and Niño 2017).

Correspondent banks provide a variety of financial services that are paramount with respect to crossborder trade. Such services include import/export credit letters, contract guarantees and discounting. If banks and other financial institutions stop providing these services, this will have an effect on the balance of payments of many countries since small and medium-sized exporters will be less able to engage in trade (Langthaler and Niño 2017). The termination of services provided by banks has the largest negative impact on small and medium-sized firms in poor countries (IFC 2017).

### 5.5.3. Increased Financial Exclusion

The implementation of a derisking strategy not only increases financial exclusion but also negatively impacts financial inclusion (Durner and Shetret 2015) and puts the world's financial integrity at risk (Babe 2017). Since financial inclusion and financial integrity complement each other, it is paramount that goals related to financial inclusion are met so that risks are mitigated in an effective manner and, more importantly, financial crimes can be combatted (World Bank 2016).

If a country or existing banked population is derisked, this means that the access of such country or existing banked population to financial services becomes curtailed (Durner and Shetret 2015). Therefore, poverty will increase, and there will be greater inequality when it comes to income (Demirgüç-Kunt and Klapper 2013). In fact, according to authors such as McLean et al. (2018), Erbenova et al. (2016), Starnes et al. (2017), and MacDonald (2019), such derisking and, particularly, the severing of CBRs also negatively impact initiatives that are primarily aimed at alleviating poverty and inequality. This is because derisking disrupts remittance flows. Additionally, people who have access to savings and credit mechanisms are in a better position to cope with economic shocks and smoothen their income (Durner and Shetret 2015).

Durner and Shetret (2015) stated that the barriers of financial inclusion include "a lack of financial literacy, low income and erratic cash flows, and high transportation and opportunity costs" (p. 21). The lack of availability of financial services is particularly detrimental in the case of developing countries, especially if there are impoverished and marginalised communities involved. Without access to crucial financial services, people are not able to purchase essential goods, pay for education or medical care and remit funds abroad (Taylor 2016).

The disruptions in global remittance flows caused by derisking is an issue of a humanitarian concern (Durner and Shetret 2015). Somalia is an example of a country that is heavily reliant on

remittances—over 40% of Somali people rely on such flows (Paul et al. 2015). A decrease in remittance flows has a major impact on the country and its most vulnerable communities (Orozco and Yansura 2013). Without such remittance flows, families would not have enough funds to pay for food, housing, education and health care (Gutale 2015).

Derisking is often perceived as being a market failure as it may adversely affect financial inclusion goals. The implementation of a derisking strategy may lead to affected clients having to rely on smaller financial institutions that do not always have sufficient resources and AML/CFT capacity to minimise the impact of risks involved. Additionally, in certain cases, higher risk clients may end up being completely pushed out of the financial sector (Nisar 2016).

*5.6. Categories of Clients Affected by Derisking*

5.6.1. Correspondent Banks

The effects of the loss of CBRs:

i.　　Trade—If the clients of banks find it difficult to send and receive foreign payments and maintain business relationships with foreign suppliers and customers, trade flows will decrease, meaning that imports and exports will also be lower than before (Haley 2018). As a result, revenues will also start to deteriorate, and businesses may find themselves unable to pay off their bank loan payments (Haley 2017).

ii.　　Banking—Weakened domestic banks may not be able to finance investment by granting loans to businesses (Haley 2017).

iii.　　Investment—If domestic banks are unable to effectively service their clients, foreign direct investment (FDI) is disincentivised. If investment expenditure starts decreasing, this may hinder a country's productivity and growth in the long-term (Haley 2017).

iv.　　Competition—Lower competition adversely affects growth and may lead to monopoly pricing, implying that social welfare can also be affected by a decrease in competition (Haley 2017).

In order to maintain its profitability, correspondent banking requires a high volume of transactions and operations (Langthaler and Niño 2017). Therefore, one of the reasons why correspondent activities are being retracted is due to this sector experiencing a decrease in returns (Langthaler and Niño 2017). Moreover, some correspondent banks are inefficiently grouping larger proportions of customers as being high risk to minimise risk and avoid potential AML/CFT scandals. This leads to such customers losing access to invaluable banking services which they require (Langthaler and Niño 2017, as affirmed by the World Bank 2015, p. 29).

Given the potential damage to reputation and balance sheet from any enforcement case, firms seek to avoid any counterparties or jurisdictions where there is uncertainty.

5.6.2. Money Service Businesses (MSBs) and Money Transfer Organisations (MTOs)

The withdrawal of correspondent banking results in derisking and directly affects MSBs, MTOs and other remittance institutions. With regards to MSBs, even if they are in full compliance with the regulations of the sending jurisdictions, transactions may still present a risk. This is not only because the jurisdictions receiving the funds sometimes have inadequate frameworks related to AML/CFT, but also due to the fact that their bordering jurisdictions may be subject to certain sanctions or have a conflict that is undergoing (Durner and Shetret 2015).

MSBs have loyal clients that regularly utilise their services. However, the amount associated with an average transaction is small. Regulatory frameworks are complex and varied, and there are additional operational and compliance burdens involved with processing a high volume of transactions. Thus, the extra costs associated with providing banking services to particular customer bases may not be offset even if there is a considerable volume of transactions (Durner and Shetret 2015).

Derisking of MTOs results in the loss of remittance services, and this can have harmful ramifications on the poor living in small countries that are not so developed when it comes to financial regulation. Examples include African, Latin American and Caribbean countries. This is because people in these countries are highly dependent on cash flows coming from abroad (Haley 2017).

This loss of income for individuals living in these countries has a negative impact on global efforts for poverty alleviation. Additionally, an increase in costs incurred to transfer money may lead to a decrease in export earnings. This results in lower investment and countries being unable to meet certain growth and development targets (Haley 2017).

### 5.6.3. Nonprofit Organisations (NPOs)

Derisking inhibits the ability of NPOs to keep on providing essential services in countries going through a humanitarian crisis, to affect aid and relief worldwide, and to campaign for changes within political and social spheres on a global scale (Goswami 2017; Durner and Shetret 2015). This especially applies to those NPOs that operate in zones experiencing conflict or instability (Durner and Shetret 2015). NPOs are perceived as being quite risky due to the nature of the work they conduct, such as transferring of money on an international level, or because the areas they operate from or both these reasons (Durner and Shetret 2015).

Although the derisking practice threatens the implementation of the mandates of humanitarian organisations and human right activists (Van Broekhoven et al. 2014), most prominent NPOs remain hesitant to speak out about the negative impacts of such practice. This is because they are afraid of even greater financial exclusion and suffering damage to their reputation (Keatinge 2014).

### 5.6.4. Other Types of Affected Clients

(i)    Charities—Charitable organisations depend on banking facilities in order to collect and receive funds (Artingstall et al. 2016). These organisations often lack the knowledge and resources needed to satisfy questions that are related to due diligence (Badour et al. 2017). The derisking process has an amplified impact on smaller charities. This is because, following a derisking exercise, smaller charities would have to start operating on a cash basis, implying more costs and higher risks (Artingstall et al. 2016).

(ii)   Politically Exposed Persons (PEPs)—Elected officials and other individuals who hold important and high profile public positions represent a higher risk due to their susceptibility to bribery and corruption. Embassies housing PEPs may also be significantly impacted by derisking (Badour et al. 2017).

(iii)  Diplomats and Foreign Students—Members of the diplomatic service of a particular jurisdiction may be discriminated against when it comes to opening a bank account or accessing financial services due to the position they hold. Additionally, foreign students may find it difficult to open bank accounts in certain jurisdictions, and this is mainly the result of problems related to identification documents and costs that may be incurred in the verification and due diligence processes (Artingstall et al. 2016).

(iv)   Defence Sector—Small and medium-sized enterprises (SMEs) specialising in the defence sector may sometimes find it difficult to obtain the required financing and access to financial services. Since several controversies related to this sector have transpired in the past, banks have started to exercise increased caution to avoid being conspicuous (Artingstall et al. 2016).

(v)    Fintech—Customers resort to this novel industry due to their inability to access, or dissatisfaction with, more traditional banking services. This industry is often the target of derisking exercises because of the high risks involved, especially in the case of virtual currency operators, electronic money institutions (EMIs) and payment institutions (PIs) (Artingstall et al. 2016; Badour et al. 2017). EMIs and PIs are extremely important because they provide services to populations that are debanked (Artingstall et al. 2016).

*5.7. The Unbanked, Vulnerable Communities, Minorities and Gender Issues*

Unbanked populations in developing countries often struggle to obtain access to financial services because of barriers that constrain their financial inclusion, such as financial illiteracy, a low level of income, inconsistent cash flows and significant costs related to transportation (Durner and Shetret 2015). The rural poor living in developing countries have historically been financially excluded by banks due to the high costs incurred to operate from remote locations (Durner and Shetret 2015).

The communities that are most significantly affected by a lack of access to financial services are "rural, low-income, and minority communities, such as women and youth" (Durner and Shetret 2015, p. 22). Women are sometimes victims of procedures that are inherently discriminatory in nature. In fact, women sometimes face financial exclusion as finance is usually much harder to obtain when compared to men, inevitably hindering women's empowerment (Durner and Shetret 2015).

*5.8. Technology-Based Solutions to the Derisking Problem*

5.8.1. Blockchain and Distributed Ledger Technologies (DLTs)

These help to mitigate the negative effects associated with derisking as their implementation helps improve the data gathering and identification processes. This ultimately leads to reduced burdens associated with compliance and enhanced transparency (Babe 2017). Moreover, blockchain and DLTs result in lower regulatory and compliance costs such as KYC requirements (IFC 2017), reduced costs associated with the verification of customers, and increased transparency of transactions.

These technology-based solutions offer numerous benefits: they facilitate the sending and receiving of remittances, make trade finance easier for businesses to obtain and assist charities operating in certain areas of conflict (Neocapita 2017).

While blockchain and DLTs are able to adequately protect confidential information, the level of anonymity provided by such technologies enables bad actors to conceal their identities. Therefore, this means that individual payments may end up being difficult to track (IFC 2017).

5.8.2. Big Data, Machine Learning and Biometrics

- Big data consists of datasets that are high in volume, velocity and variety, meaning that unlike traditional datasets, the former necessitates the use of specialised systems and analytical techniques. One of the main advantages of big data is that it permits a wide variety of data types to be stored in the same place, implying that staff will now spend less time gathering information from different sources. Big data also increases the range and scope of information that can be used when it comes to KYC and suspicious transaction investigations (Woodsome and Ramachandran 2018).
- Machine learning is essentially a form of artificial intelligence that leads to computers becoming more efficient at performing certain tasks through repeated iterations. Such new technology can be used to help transform compliance functions of organisations and also results in more accurate monitoring of transactions. This is because machine learning identifies previously undetected illicit finance techniques and reduces false alerts (Woodsome and Ramachandran 2018).
- Biometrics are used to authenticate the identity of a person and control his/her accessibility to a particular system. By utilising biometrics, banks are able to identify and verify the identities of their customers and carry out due diligence in a proficient manner. This addresses gaps related to identification that are present in developing countries (Woodsome and Ramachandran 2018).

*5.9. Derisking in the Context of Malta*

5.9.1. The MONEYVAL Report of 2019

In 2019, MONEYVAL, a monitoring body of the Council of Europe (CoE) specialising in antimoney laundering, published a report that provided a brief overview of the AML/CFT measures Malta has implemented and commented on the robustness of the AML/CFT system the country has in place.

Moreover, this report analysed the extent to which Malta is compliant with the FATF recommendations (CoE 2019a).

To obtain a more comprehensive understanding of money laundering and financing of terrorism risks, Malta conducted a national risk assessment (NRA) exercise in 2013/2014. Following this NRA exercise, Maltese authorities were able to more extensively comprehend the risks and vulnerabilities in Malta's AML/CFT system, especially in the case of regulated sectors (CoE 2019a). However, MONEYVAL noted that there were factors, including predicate offences, legal persons and arrangements and financing of terrorism, which appeared to be insufficiently understood (CoE 2019b).

### 5.9.2. Derisking in Malta: A Local Case

The banking sector in Malta has been experiencing a downward trend for these past few years. Late in the year 2018, Pilatus Bank, which was considered Malta's most controversial bank, had to close down its operations after allegations of money laundering breaches emerged. In July 2019, Satabank was fined a sum of €3 million for having weak structures in place, which facilitated the carrying out of criminal activities such as money laundering (Vento 2019).

Lately, Malta's banking sector has increasingly suffered as foreign banks are withdrawing their correspondent bank services from Malta (Costa 2019, Costa 2019). This has mainly been the result of failures vis-à-vis antimoney laundering in Europe (Vento 2019).

In the year 2019, a core domestic bank, Bank of Valletta (BOV), embarked on a derisking exercise (Grech 2019). This resulted in BOV ceasing the provision of certain banking services to some of its customers and closing their bank accounts (Martin 2019). To ensure that it followed international best practices, BOV decided to undertake this derisking exercise alongside a KYC retail client remediation exercise (Grech 2019).

The decision to initiate such derisking was taken following discussions BOV had with the Malta Financial Services Authority (MFSA) and the European Central Bank (ECB) (Grech 2019). It was brought to light that the ECB had given BOV clear instructions to engage in derisking in order to establish a new risk profile that involves decreased risk (Martin 2019). Derisking was also carried out so as to allow BOV to strengthen its frameworks, systems, policies and procedures regarding risk, compliance, antifinancial crime and antimoney laundering (Costa 2019).

The customers affected by this derisking exercise mainly comprised of businesses deemed as being high risk, such as iGaming companies. These iGaming companies, including Malta Gaming Authority (MGA) licensees, had their bank accounts closed (Martin 2019). Other affected customers that also had their accounts closed consisted of gaming affiliate companies (Martin 2019), as well as individual investor programme (IIP) buyers who had opened a bank account in Malta to obtain a Maltese passport (Grech 2019).

Derisking was carried out with respect to both international corporate customers (ICC) and international personal banking (IPB) customers. ICCs typically consist of companies that are registered in Malta with the Malta Business Registry (MBR), where the majority of the company's business activity occurs outside of Malta, and the beneficial owner is not Maltese (Grech 2019). When it comes to IPB customers, there is a mix of clients, with some located outside EU jurisdictions. The accounts of ICCs and IPB customers that BOV determined as not having an economic nexus in Malta were closed (Grech 2019).

The implementation of this derisking strategy has made it difficult and cumbersome for foreign companies to set up their operations in Malta and open a bank account (Martin 2019). Additionally, given limited local options, a number of foreign clients had no other choice but to transfer their operations overseas or start making use of e-money institutions (Vento 2019).

## 6. Methodology

### 6.1. Tool Used to Collect Data

For the purpose of this study, a purpose-built semistructured survey was designed and administered to personnel in banks, financial institutions, accounting firms, regulatory authorities, as well as any other relevant parties and people dealing with these institutions.

Some of these surveys, which were built using a software application (Qualtrics XM) in order to provide an online link, were administered using social media and email; the rest were carried out over the phone or using other communication applications such as Skype.

The survey consisted of three sections. The first section consisted of five demographic questions asking for the "age", "gender", "locality", "capacity" and "level of expertise" of the participant. The next section consisted of six themes with three statements each (i.e., 18 statements in total), to which participants were asked to rate each statement using a 1 to 5 Likert scale, "1" being "Strongly Disagree" and "5" being "Strongly Agree". The themes were associated with certain drivers, factors and implications of derisking identified from existing literature (derived using the thematic approach, as suggested by Braun and Clarke 2006). These themes were "Financial", "Socioeconomic", "Financial Exclusion", "Antimoney Laundering/Combatting the Financing of Terrorism (AML/CFT) and Risk", "Compliance and Regulation" and "Technology-Based Solutions". The third section consisted of a set of seven open-ended questions related to the topic of derisking and its main drivers, factors, implications and effects, particularly with regards to the Maltese scenario (Saunders et al. 2009; Yin 2002; Yazan 2015; Stake 1995). Other questions dealt with

- The advantages and disadvantages of derisking;
- Potential solutions to the main derisking problems;
- The role of regulators and regulatory authorities vis-à-vis derisking;
- The Maltese sectors and industries that are the most negatively affected by derisking; and
- The prevalence of derisking in Malta in the years to come.

### 6.2. Sample Population

Since the population of possible candidates was not known, we used a mix of nonprobability purposive sampling by contacting persons on our social media and email, and they connected us with further possible participants (nonprobability snowballing sampling; Mack et al. 2005; Neuman 2005).

The initial sample of potential study participants consisted of 60 organisations employing individuals who possess a relatively high level of knowledge regarding the derisking practice, money laundering and terrorism financing. The organisations that were targeted included banks operating in Malta, the Central Bank of Malta (CBM), local regulatory authorities (the MFSA and the FIAU), the large accounting firms and other accounting and risk management firms.

To determine the participants' level of expertise, a 1 to 3 Likert scale was utilised, where 1 corresponded to "a fair level of expertise", 2 corresponded to "a good level of expertise" and 3 corresponded to "an excellent level of expertise". To ensure that the data obtained were reflective of expert knowledge, we filtered participants by level of expertise and only used data from participants who expressed a good or excellent level of expertise.

Between 15 November 2019 and 21 April 2020, we carried out a total of 32 interviews with individuals who have a good or excellent level of expertise in derisking. Although it seemed that additional interviews beyond this point did not provide new information or added value, we maintained an open mind and supplemented our data with the 296 completed online surveys received from participants during the same period in which we carried out the interviews. This ensured that our selection of participants was, as much as possible, free from any selection bias since the surveys were answered anonymously. These surveys were then filtered to ensure that the level of expertise criteria was met, and these resulted in 285 valid participants. In total, we had 317 participant surveys.

*6.3. Analysis*

Following the collection of data, such data was first transferred into Microsoft Excel and Microsoft Word, and, later, the quantitative data were loaded onto Statistical Package for the Social Sciences (SPSS version 21) software. Subsequently, the data were analysed using frequencies and by subjecting the data responses to the statements to exploratory factor analysis, which is a method of testing the theoretical understanding of the factors for the main drivers of derisking posed in the first research question (RQ1), and, as determined from the literature, no a-priori fixed number of factors was set. The final number of factors was determined by the data and the authors' interpretation of them (Hair et al. 1998).

Since the items used the ordinal scale of measurement, we used the median (Md) as a measure of central tendency and the interquartile range (IQR) as a measure of spread. Where a group of items could be grouped into a construct (or theme), we assessed the internal consistency reliability of the measures via Cronbach's alpha. After the items were combined into a single Likert scale, we computed the mean (M) as a measure of central tendency and the standard deviation (SD) as a measure of spread.

We then carried out multiple linear regression to determine how this measure varies with the demographic factors; we also used STATA application software applying White robust standard errors to account for any heteroskedasticity and analysed the answers to the last section using the thematic analysis approach, as suggested by Braun and Clarke (2006, p. 6), to answer RQ2.

## 7. Results and Discussion

*7.1. Summary Statistics*

As can be noted from the tables in Appendix B, the largest group of participants were aged between 30 and 39 (39.7%), most were men (68.5%), coming from a central location (54.9%), and users of financial institutions (67.5%), with a good level of expertise (59.6%). This shows that most of the participants were exposed to a similar culture and environment, and this could be the reason why the results of the multiple linear regression, noted and discussed below, are not significant.

Additionally, as noted in Appendix A, all participants are in agreement (mean between 3.4 and 4.19) with the statements posed in the survey/questionnaire, with the exception of Statement 13 (S13) wherein the participants strongly agreed. This further confirms our findings from the multiple linear regression carried out and explained below.

*7.2. Exploratory Factor Analysis*

For exploratory factor analysis, we used direct oblimin via principal components extraction with Kaiser normalization. The Kaiser–Meyer–Olkin (KMO) statistic, which is a measure of sampling adequacy for the appropriateness of applying factor analysis, fell within the acceptable range (above 0.6) with a value of 0.93. This further supported the continuance of factor analysis, and so the analysis proceeded.

Exploratory factor analysis loaded best as three factors and 22 statements, which, in combination, explained 60.68% of the variance of the perceived implications of derisking in Malta. Table 2 shows which statements are grouped under each of the three factors. Factor 1, which has now been termed "Research and Development Solutions", explained 46.687% of the variance and comprised five items that include the following statements:

S3    Derisking leads to an increase in costs incurred to transfer money. Therefore, export earnings decrease, resulting in a lower level of investment. Moreover, given that domestic banks will not be able to effectively service their clients, this leads to a lower level of foreign direct investment (FDI). As a result, a country's productivity and growth in the long-term may be negatively affected, rendering such a country unable to meet growth and development targets.

S5　　The implementation of a derisking strategy increases financial exclusion and negatively impacts financial inclusion. If a country or existing banked population is derisked, this means that the access of such country or banked population to financial services becomes limited. When access to financial services is restricted, poverty and income inequality increase.

S16　In order to mitigate the negative effects associated with derisking, certain technology-based solutions can be implemented. These include blockchain, distributed ledger technologies (DLTs) and biometrics. The implementation of such solutions improves data gathering and identification processes, ultimately leading to reduced burdens associated with compliance and enhanced transparency.

S17　Blockchain and DLTs help minimise the negative impacts of derisking by facilitating the sending and receiving of remittances, making trade finance easier for businesses to obtain and assisting charities operating in certain conflict areas.

S18　While blockchain and DLTs are able to adequately protect confidential information, the level of anonymity such technologies allow makes it easy for bad actors to conceal their identities. Therefore, if this happens, individual payments become more difficult to track.

**Table 2.** Exploratory factor analysis [a].

|  | **Factors** | | |
| --- | --- | --- | --- |
|  | **1** | **2** | **3** |
| S17 | 0.857 |  |  |
| S16 | 0.803 |  |  |
| S18 | 0.791 |  |  |
| S7 | 0.719 |  |  |
| S6 | 0.546 |  |  |
| S14 |  | 0.812 |  |
| S1 |  | 0.697 |  |
| S5 |  | 0.675 |  |
| S3 |  | 0.564 |  |
| S13 |  | 0.519 |  |
| S15 |  | 0.496 |  |
| S12 |  | 0.404 |  |
| S8 |  |  | −0.794 |
| S10 |  |  | −0.759 |
| S9 |  |  | −0.741 |
| S4 |  |  | −0.620 |
| S11 |  |  | −0.448 |
| S2 |  |  | −0.371 |

Extraction method: principal component analysis. Rotation method: oblimin with Kaiser normalization. [a] Rotation converged in 12 iterations. Source: authors' computations.

Factor 2, which has now been termed "Compliance and Regulatory", explained 7.621% of the total variance and comprised seven items that include the following statements:

S1　At times, banks and other financial institutions may opt to terminate certain correspondent banking relationships (CBRs) with smaller foreign banks and close their correspondent bank accounts, consequently limiting their access to foreign currencies. Therefore, the derisking process leads to a reduction in international trade finance.

S3　Low profitability of customer bases also brings about derisking. If the profitability of a particular customer base does not compensate for the higher risks and additional costs that must be incurred, such a customer base is often derisked. An example of these additional costs is Know Your Customers' Customers (KYCC), which means that banks and other financial institutions are not only responsible for their customers, but also for the customers of their customers.

S5   The derisking practice leads to lower imports and exports as the customers of banks engaging in derisking will not be able to send or receive foreign payments and maintain important business relationships with foreign suppliers and customers.

S12  In response to derisking, many customers are resorting to shadow banking channels and fintech to obtain required funds. These environments are unknown and much less regulated and monitored, implying that risk is higher as well. If such alternatives are the only source of finance available, this means that consumers may have to incur higher costs to address their financing needs.

S13  With AML/CFT regulations always increasing and becoming ever more complex, this has resulted in a rise in compliance costs. These increased compliance costs have a magnified effect on jurisdictions that have a smaller and more restricted banking market. This is because such jurisdictions are often unable to spread the fixed cost component of these AML/CFT compliance costs over a large enough number of transactions.

S14  Since the fines and penalties imposed for noncompliance with and breaches of AML/CFT requirements are considerable, many banks and other financial institutions have opted to implement a derisking strategy.

S15  The imposition of increased capital and liquidity requirements following the 2008 global financial crisis is also another factor which is influencing the derisking carried out by banks and other financial institutions.

Factor 3, which has now been termed "Direct and Indirect Effects", explained 6.373% of the total variance and comprised six items that include the following statements:

S2   Correspondent banks provide a variety of financial services that are paramount with respect to crossborder trade. Such services include import/export credit letters, contract guarantees and discounting. If banks and other financial institutions cease providing these services, small and medium-sized exporters will be less able to participate in trade. This especially applies to small and medium-sized firms in poorer countries.

S4   The withdrawal of CBRs and derisking of money transfer organisations (MTOs) result in the loss of remittance services. This can have negative ramifications for the poor living in small countries that are not so developed when it comes to financial regulation, e.g., African, Latin American and Caribbean countries.

S8   The lack of availability of financial services is particularly detrimental in the case of developing countries, especially if there are impoverished and marginalised communities involved. Without access to financial services, people would not be able to purchase essential goods, pay for education or medical care, and remit funds abroad.

S9   Derisking has a significant negative impact on nonprofit organisations (NPOs) as it inhibits their ability to provide crucial services in countries currently undergoing humanitarian crises, affect aid and relief, and campaign for changes in political and social spheres.

S10  Following the 2008 global financial crisis, banks and other financial institutions worldwide started to curtail their risk appetite and adopted a risk-based approach (RBA) with regards to AML/CFT. This reduction in the risk appetite has led to an increase in derisking.

S11  When banks and other financial institutions engage in derisking, this results in a shift in AML/CFT risk. This is because if CBRs are terminated, the affected clients will then have to start relying on smaller banks and credit institutions to obtain the financial services they require. The problem with this is that such smaller banks and credit institutions may not have the expertise and capacity necessary to service high-risk and poor populations.

The 18 statements can be used as an inventory checklist grouped under the 3 main themes "Research and Development Solutions", "Compliance and Regulatory" and "Direct and Indirect Effects" to understand the implications that a derisking exercise can bring to a financial firm. Moreover, one can use this checklist/inventory to develop a risk exposure matrix to arrive at a single qualitative rating and/or quantitative measure of the implication of derisking.

This is of specific interest to risk managers working in the financial services sector, who, as per regulatory requirements, would need to summarise and classify the risk rating of a firm, project and strategy to the board of directors, shareholders and the regulators. It is also of interest to policymakers who can develop policies based on such results.

### 7.3. Reliability Test

The Cronbach's alpha coefficients of this scale were between 0.83 and 0.86. Therefore, we can conclude that this scale is reliable as part of our statistical analysis (Table 3).

**Table 3.** Cronbach's alpha values (*n* = 317).

| Factor | Item | Mean | Min–Max | Cronbach's Alpha |
|--------|------|------|---------|------------------|
| 1 | 5 | 4.09 | 3.97–4.27 | 0.88 |
| 2 | 7 | 3.91 | 3.80–4.21 | 0.83 |
| 3 | 6 | 4.09 | 3.88–4.29 | 0.86 |

Source: authors' computations.

The computed "Derisking Model" for Malta shows a mean of 4.03 (SD = 0.67). All the factors (Factors 1, 2 and 3) produced means that were close to the computed model. This shows that participants from Malta, overall, believe that these are the drivers of derisking in Malta.

### 7.4. Multiple Linear Regression and ANOVA

The computed one-way analysis of variance (ANOVA) was used to show that there are no statistically significant differences between the means of the independent (unrelated) groups (*p* > 0.01; Table 4).

**Table 4.** ANOVA [a].

| | Model | Sum of Squares | df | Mean Square | F | Sig. |
|---|-------|----------------|-----|-------------|---|------|
| 1 | Regression | 2.371 | 5 | 0.474 | 1.043 | 0.392 [b] |
| | Residual | 141.380 | 311 | 0.455 | | |
| | Total | 143.751 | 316 | | | |

[a] Dependent variable: derisking drivers. [b] Predictors (constant): level of expertise, age, provider–user, gender, and locality. Source: authors' computations.

Moreover, multiple regression analysis (*p* > 0.01) reveals that the perception of participants and interviewees does not change as an effect of the different demographics, i.e., (1) age, (2) gender, (3) locality, (4) whether they are a provider or a user of financial services, and (5) level of expertise (Table 5). The same result is obtained after following the use of White robust standard errors to account for heteroscedasticity (Table 6). This shows that all participants are aligned in their opinion and reasoning about the main drivers of a derisking strategy, and these opinions are not dependent on any demographic factors. This may be due to the fact that Malta is a small island and one of the small European Union jurisdictions with a population of approximately 480k in an area of 246 km squared, with one main national university, a single financial services regulator (Malta Financial Services Regulator (MFSA)) and a Financial Intelligence Analysis Unit (FIAU). Therefore, most of the participants have practiced, worked and studied in/under the same environment or are in contact at similar conferences.

**Table 5.** Coefficients [a].

| Model | | Unstandardized Coefficients | | Standardized Coefficients | t | Sig. |
|---|---|---|---|---|---|---|
| | | **B** | **Std. Error** | **Beta** | | |
| | (Constant) | 3.868 | 0.313 | | 12.341 | 0 |
| | Age | 0.008 | 0.045 | 0.011 | 0.169 | 0.866 |
| | Gender | −0.102 | 0.095 | −0.07 | −1.074 | 0.284 |
| 1 | Locality | 0.058 | 0.07 | 0.056 | 0.832 | 0.406 |
| | Provider-User | −0.111 | 0.085 | −0.077 | −1.302 | 0.194 |
| | Level of Expertise | 0.128 | 0.088 | 0.093 | 1.451 | 0.148 |

[a] Dependent variable: derisking drivers. Source: authors' computations.

**Table 6.** Accounting for Heteroskedasticity.

| Variable | Coefficient |
|---|---|
| Age | 0.0075 (0.0401) |
| Gender | −0.1025 (0.0991) |
| Locality | 0.0579 (0.062) |
| Provider-User | −0.1108 (0.0849) |
| Level of Expertise | 0.1269 (0.084) |
| Constant | 3.8713 *** (0.2937) |
| N | 317 |
| R-Squared | 0.0165 |
| F-Statistic | 1.08 |

Notes: Dependent variable is the score denoting derisking drivers. Robust standard errors are reported in parentheses. *** denotes statistical significance at the 1% level; ** denotes statistical significance at the 5% level; * denotes statistical significance at the 10% level. Source: authors' computations.

## 7.5. Solutions to the Main Problems of Derisking (Participants Responses to the Interviews And/or Open-Ended Questions)

With respect to mitigating the negative effects of derisking, emerging technology such as blockchain and distributed ledger technology (DLT) were the two most common solutions provided by the participants (35). If coupled with a "robust regulatory environment and governance structure", blockchain and DLTs have the potential to reduce the costs and burdens of regulation and compliance and also increase the overall level of transparency. This is in agreement with the views of Babe (2017) and the IFC (2017).

Nevertheless, some participants (21%) emphasised the fact that blockchain and DLTs are still in their "infancy stage" and not widely accepted yet. In fact, they are by no means a "silver bullet" and there are still considerable limitations related to these technologies that need to be sorted out. Firstly, blockchain and DLTs depend a great deal on other technologies such as the internet, which is not always available in certain geographical areas, particularly in developing countries and other countries perceived as being high risk. In addition, these technologies lack scalability, and the speed of transactions using such technologies is considered as being slower compared to transactions being processed using existing technologies. The underlying concept of decentralisation conflicts with

regulatory objectives as it makes it difficult to determine the controlling parties. Finally, there are also issues related to data protection.

One participant stated that while "blockchain and DLTs are tools that can be both beneficial and detrimental" since these technologies are designed to build a system of trust amongst a group of individuals, they can be abused if such technologies are used for the wrong reasons. A further argument brought up by another interviewee is that these technologies are usually considered as being ancillary tools and not essential tools for banks and other financial institutions. This is because of the high costs associated with implementing and monitoring the use of such technologies. One interviewee was completely against the use of blockchain and DLTs and argued that these technologies are not mature enough, and there are too many hurdles which need to be surpassed for them to become mainstream.

Alternative solutions to the problems of derisking suggested by a few interviewees (13) include legal entity identifiers (LEIs) and Know Your Customer (KYC) utilities. LEIs are based on International Standards Organisation (ISO) 17,442 and consist of a code that is made up of 20 alphanumeric characters. These identifiers allow legal entities participating in financial transactions to be clearly and uniquely identified, leading to enhanced transparency on a global scale. KYC utilities enable financial institutions to effectively and efficiently carry out their KYC procedures through the centralised collection, verification and sharing of customer information.

Another participant believed that a possible solution to mitigate the adverse impacts of derisking strategies implemented in Malta is to have a financial services sector that constitutes a large number of small players, each having a different and varied risk appetite, specialising in the management of customer relationships characterised with their own unique risk level. Naturally, the management of customers that present a higher risk attracts higher fees for the services provided. Blockchain, DLTs and fintech adopted in a regulated manner "are all key in this multiplayer landscape of financial services providers".

The introduction of stricter regulation and AML/CFT compliance requirements means that enhanced oversight by the ECB and Maltese regulatory authorities, such as the MFSA and the Financial Intelligence Analysis Unit (FIAU), is now more crucial than ever. All interviewees (32) advocated that regulators and regulatory authorities are responsible for issuing robust guidance to banks and financial institutions, not only in the case of Malta but also internationally. This thereby ensures that an appropriate RBA is applied and that controls are administered when there are high-risk customers involved. One interviewee noted that although oversight by local and international regulators and regulatory authorities has been increasing steadily, such oversight does not always take into consideration the principle of proportionality. The example given by this interviewee was that a Maltese bank and German bank might be treated in the same way, "notwithstanding the obvious difference in size and systemic importance". To ensure fairness and equality, it is, therefore, paramount that regulators and regulatory authorities apply this principle well.

Another interviewee expressed that regulators and regulatory authorities must ensure that meaningful derisking exercises are executed. This implies that the sufficient controls that allow the appropriate management of risks need to be in place so that residual risks are kept to a minimum. This interviewee then added that these controls should be continuously monitored "so that their effectiveness is maximised". It is also very important that regulators and regulatory authorities ascertain that the customer bases held by banks and other financial institutions are sustainable.

In the scenario of Malta, it seems to be that local regulatory authorities are the ones initiating and imposing derisking exercises themselves, rather than the Maltese banks and other financial institutions being the ones voluntarily engaging in derisking. One of the interviewees believed that this approach is being taken because such regulatory authorities want to make sure that local banks and other financial institutions understand their business models well and manage the risks involved. This leads to "an increase in the level of risk awareness across the general society".

One interviewee believed that after the MONEYVAL Report was issued in 2019, Maltese regulatory authorities shifted their focus and changed their role from being watchdogs to "blood hounds".

However, this interviewee feared that the introduction of more stringent requirements and higher fines and penalties will not solve the money laundering and terrorism financing issue with regards to the Maltese situation and contended that different strategies will need to be explored in the long-term.

Twelve interviewees explained that local regulatory authorities play an important role when it comes to safeguarding the integrity and stability of Malta's financial system. Their role also ensures that the recommendations of prominent regulatory bodies, such as the Basel Committee on Banking Supervision (BCBS) and the FATF, are adhered to as this "helps to greatly improve the country's international standing from a compliance perspective". Consequently, compliance costs for correspondent banks would decrease as such banks would start viewing Malta as presenting a lower risk, implying that less money and resources are required for the monitoring of relationships.

While strict supervisory mechanisms are a fundamental component that helps build and restore trust in banks and other financial institutions, it is of utmost importance that local and international regulators and regulatory authorities do not overregulate the market as this may force certain customer bases into less regulated channels. According to one interviewee, the "right balance which adequately manages risk, while not extinguishing business, needs to be struck". Another interviewee added that regulators and regulatory authorities "need to be careful so as to not overburden banks and financial institutions with too much bureaucracy to the point where it then becomes overkill".

### 7.6. Derisking within the Maltese Context

The NRA that was carried out by the Maltese authorities in 2013/2014 identified certain sectors and industries that were the most adversely impacted by derisking undertaken by the local banks and financial institutions. Such sectors and industries, together with other sectors that the general public has a negative perception of, include the iGaming sector, companies which provide services that are related to virtual financial assets and passport by investment scheme promoters.

Almost all of the interviewees (29) remarked that the majority of gaming companies in Malta are being derisked by local banks, even if such companies are licensed by the MGA. This is because of the high risk that these companies carry. This ties in with the literature; as part of their derisking strategy, Maltese banks sometimes decide to cease providing certain services to gaming companies and close their bank accounts. One interviewee stated that such derisking by banks and financial institutions is a product of the "extended regulatory scrutiny that Malta has been under due to being open to high-risk sectors with limited enforcement".

Companies that provide services in the area of virtual financial assets, such as cryptoassets and DLT assets, may find it difficult to find a bank that is willing to offer them traditional banking services. This is a result of the unknown risks that are associated with new and innovative technologies like cryptocurrencies and DLTs.

The Maltese banking and financial services sectors are also considerably impacted by the derisking process. Several interviewees (22) expressed that it is not easy for local banks and other financial institutions to establish CBRs with foreign banks and financial institutions, which leads to lower activity in business lines such as the "provision of payments in foreign currencies and trade finance". This, in turn, affects the business generation capabilities of companies that have a material proportion of their revenues generated from international trade. A lack of a proper structure in place that facilitates foreign currency transactions impedes business from being conducted.

Other Maltese sectors, industries and organisation types that interviewees (28) noted as being relatively high risk from a money laundering and terrorism financing perspective include fintech, tourism (incorporating hospitality and catering), shipping, property development, certified financial planners offering trustee and fiduciary services, company service providers, trading companies and MSBs.

*7.7. Will Derisking Become More Prevalent in Malta in the Years to Come?*

All the interviewees, except for one, claimed that in the near future, derisking in Malta will continue to be undertaken by banks and other financial institutions. First of all, signs in the global economy are indicating that a step back from globalisation levels achieved so far seems to be desirable. Consequently, it is possible that this will lead to further derisking, not only in Malta but also on an international level. When taking into consideration the local context, many interviewees (22) stated that the drivers of derisking are ever-increasing regulatory pressures and compliance requirements imposed by the ECB and Maltese authorities. As a result, these interviewees predicted that these drivers, coupled with all the recent developments in AML/CFT legislation, will result in the further exercise of the derisking practice in Malta.

According to one interviewee, it is inevitable that derisking strategies will continue being implemented in the short- to medium-term as pressure keeps on mounting for Maltese authorities and institutions to "come clean". However, in the long-term, this interviewee believed that business opportunities might open up for institutions that are smaller in size in the form of customers that will end up unbanked by larger institutions. If these smaller institutions obtain a comprehensive and thorough understanding of the inherent risks associated with banking these customer bases, they would be able to offer their services and make a decent return.

In line with the findings of the MONEYVAL Report issued in 2019, local banks and other financial institutions are expected to continue engaging in derisking so that existing AML/CFT deficiencies can be addressed and Malta's standing can improve. This sentiment was conveyed by five interviewees, who also added that continued and ongoing derisking by Maltese institutions will make it easier for them to maintain business relationships with foreign institutions and will decrease compliance risk.

Other interviewees (12) contended that in the years to come, derisking exercises in Malta will start being carried out not only by banks and other financial institutions but also by gaming, insurance, property development, asset management and wealth management companies.

Several interviewees also discussed derisking in Malta within the context of the recent COVID-19 outbreak. Currently, in the midst of this unprecedented pandemic, Maltese authorities are trying to relax and alleviate regulatory requirements so that development and growth are not stifled. One of the interviewees commented that "since everything is covered by a shroud of uncertainty", it is highly possible that regulatory requirements may be relaxed for a prolonged period of time. This is due to the fact that post-COVID 19, such relaxed requirements would serve a stimulus for banks and financial institutions to start regenerating the economy.

A final interesting argument brought forward by another interviewee is the fact that although derisking is unavoidable in the current environment, a point of saturation may eventually be reached, and this would probably end up stalling the entire sector. This would then be followed by a "new cycle of easing of regulations", with the aim of achieving a new equilibrium.

## 8. Conclusions

Derisking is a global phenomenon that has recently been gaining traction both locally and internationally. In the case of Malta, derisking is still an emerging practice that is not very well-established, which has a lot of potential for further growth and development. In fact, it is only during the past few years that Maltese banks and other financial institutions have started undertaking derisking exercises to better manage their risks, focus on customer bases that fall within their risk appetite and ensure compliance with regulatory and AML/CFT requirements.

There are many factors that are driving derisking within the Maltese scenario. Firstly, a considerable amount of EU directives and regulations, especially those regarding AML/CFT, have been introduced. Since the costs of compliance associated with such directives and regulations are relatively high, this has led to the increased prevalence of derisking both in Malta and the EU.

Locally, regulatory pressures from the ECB and local regulatory authorities on Maltese banks and other financial institutions have been significantly intensifying. Pressure has been steadily mounting

for these local banks and financial institutions to re-evaluate their risk profiles and enhance their strategies related to AML/CFT. Consequently, such banks and financial institutions have no choice but to derisk certain riskier customer bases.

The MONEYVAL report, issued in 2019, is another factor leading to derisking in Malta; this report has had a two-fold effect. Firstly, the report calls on Maltese authorities to improve the AML/CFT measures the country has in place, and this has, in turn, led to the authorities compelling local banks and other financial institutions to decrease their risk appetite, resulting in higher derisking. Secondly, given that the report pinpoints several AML/CFT deficiencies present in Malta, foreign banks and financial institutions have started to exercise more caution and sometimes restrict the services provided to, or simply exit relationships with, certain local banks and financial institutions.

Reputational risk is also leading to increased derisking in Malta. The imposition of higher fines and penalties by regulators on banks and other financial institutions for noncompliance with AML/CFT requirements, and the adverse media brought about by such fines and penalties, is leading to a decline in the risk appetite of local banks and financial institutions. This ultimately results in the derisking of customer bases. Malta's current turbulent political climate and the many scandals and allegations of corruption that were recently brought to light have led to a deterioration in the country's reputation. As a result, foreign correspondent banks have initiated the severing of relationships with some Malta banks and other financial institutions because the significant AML/CFT compliance costs are not justified by the high risk and low volume of transactions.

Following the 2008 global financial crisis, capital and liquidity requirements have considerably increased. Banks now need to hold larger capital buffers so that they are able to decrease the risks of their portfolios. This has had a negative effect on the profitability of both local banks as well as banks within the EU and is leading to more widespread derisking. In fact, some relationships with higher risk customers that are not that profitable to maintain are being terminated.

With regards to the general implications and effects of the derisking process, the interviewees mentioned a shift in AML/CFT risk. After being derisked, customers have to resort to other smaller banks and financial institutions to obtain the financial services they require, even though such banks and financial institutions may lack the capacity needed to service such customers. Other implications that the interviewees highlighted were the use of shadow banking channels to acquire finance, the loss of legitimate business and the financial exclusion of certain customer bases.

The interviewees explained that derisking can be seen from two perspectives. From the perspective of banks and financial institutions engaging in derisking, adopting this practice lowers their risk exposures, reduces the risk of fines and penalties, decreases compliance costs and minimises reputational risk. From the perspective of financial markets, derisking safeguards the interests of lower-risk customers and enhances the integrity of the financial system.

Since Malta is one of the smallest countries in the EU, this implies that a robust banking sector is vital in order to attract top-level players to the island. The country's shift from a sales-oriented mentality to a compliance and regulatory mentality signifies that more derisking exercises need to be introduced. The implementation of derisking strategies also ensures that the reputational risk of Maltese banks and other financial institutions is kept to a minimum.

The undertaking of increased derisking in Malta should lead to less corruption and money being leaked from the economy due to money laundering, meaning that trust in the local banking and financial services sectors should also increase. By adjusting their risk appetites, local banks and other financial institutions can contribute to the rebuilding of Malta's reputation. Furthermore, by engaging in derisking, such banks and financial institutions will shrink and become more stable, implying that additional attention can be directed towards core customers.

Apart from its benefits, derisking also has its drawbacks. Derisking exercises may result in a deceleration of economic activity, and this has an impact on the whole socioeconomic system, leading to a decrease in GDP, employment levels and economic growth. Subsequently, Maltese institutions and organisations may have to downsize their operations, and some employees may end up redundant.

When asked about whether the advantages of derisking outweigh the disadvantages, especially in relation to the Maltese scenario, the responses of interviewees were mixed. Some interviewees argued that proper implementation of derisking achieves more pros than cons as it restores the balance between profitability and having customers that are of high quality. In the short-term, the derisking practice leads to a decrease in the profits of banks and other financial institutions, but in the long-term, profits would be more sustainable.

Other interviewees stated that although the adoption of derisking can result in lower compliance costs and higher efficiency, a lack of proper understanding and capacity can result in the loss of legitimate business, defeating the purpose of this exercise. Additionally, the many drawbacks of the derisking process, such as declining CBRs, the use of unregulated banking channels and financial exclusion, can render derisking unsustainable. The solution provided to maximise the pros and minimise the cons of derisking is to find the right balance and mechanism. Regulation should also be balanced and proportionate.

When it comes to the Maltese scenario, given all the recent AML/CFT developments and regulatory pressures by the ECB and local authorities, derisking is on the rise and expected to continue increasing in the near future. Maltese regulatory authorities have the important role of not only providing oversight and supervision but also of facilitating the undertaking of meaningful derisking. This ensures that local banks and financial institutions are able to effectively manage their risks and have customer bases which are sustainable.

In conclusion, the objective of this study, i.e., to identify and analyse the factors affected by and the implications of the derisking process, has been achieved, as clearly shown in the summary above. The research questions that examine the drivers and implications of derisking in the case of Malta have also been answered and exhaustively analysed and discussed throughout this study.

In light of the recent scandals and corruption allegations, the Maltese authorities need to, now more than ever, strengthen the country's fight with regards to money laundering and terrorism financing. One way that this can be achieved is by the imposition of derisking exercises on other sectors of the economy.

The derisking approach should be standardised across the board to ensure a level playing field for all stakeholders. If all Maltese institutions implementing a derisking strategy adopt certain common criteria and standards, this will lead to enhanced transparency and less discriminatory practices applied. Standardisation of the derisking process ensures that the derisking that is conducted is more meaningful and only the customer bases that pose the greatest risk are derisked. This standardised approach should result in more fairness and consistency, therefore maximising the effectiveness that can be derived from undertaking derisking.

Furthermore, the Maltese regulatory authorities should be provided with adequate human and financial resources as this allows them to have the capacity required to carry out enhanced oversight and risk-based supervision based on the size, complexity and risk profile of Malta's private sector. This would also permit such authorities to introduce more practical AML/CFT measures and carry out more investigations and prosecutions. Moreover, by having sufficient resources, the authorities would be able to monitor those Maltese institutions undergoing derisking and ensure that the process is being properly and equitably conducted. Law enforcement authorities should, similarly, have the resources necessary to pursue high-level cases of money laundering, bribery and corruption.

More education and workshops regarding this practice should also be provided to professionals working within the fields of banking, financial services and accountancy. Moreover, it is important that such professionals keep up-to-date with the latest changes and developments when it comes to directives and regulations concerning AML/CFT, as well as derisking in general.

**Author Contributions:** J.M.A. and S.G. contributed towards the conceptualization, methodology, software, validation, formal analysis, investigation, and resources. J.M.A. took care of the data curation, and writing—original draft preparation. S.G. carried out the writing—review and editing, visualization, supervision, and project administration. P.J.B. carried out the final editing, analysis, and review. J.V.S. carried out the final editing, analysis,

and review. "The Implications of Derisking: The Case of Malta, a Small EU State". All authors have read and agreed to the published version of the manuscript.

**Funding:** This research received no external funding.

**Acknowledgments:** This paper is based on an unpublished thesis as the final part for the MA in Accountancy at the Accountancy Department, University of Malta, by J.M.A. (2019). "The Implications of Derisking: The Case of Malta" was supervised by Simon Grima—herein the main authors.

**Conflicts of Interest:** The authors declare no conflict of interest.

## Appendix A

**Table A1.** Summary statistics.

|  | N | Minimum | Maximum | Mean | Std. Deviation |
|---|---|---|---|---|---|
| S1 | 317 | 1.00 | 5.00 | 3.8013 | 0.96206 |
| S2 | 317 | 1.00 | 5.00 | 4.1956 | 0.88924 |
| S3 | 317 | 1.00 | 5.00 | 3.9685 | 0.98996 |
| S4 | 317 | 1.00 | 5.00 | 4.1041 | 0.92361 |
| S5 | 317 | 1.00 | 5.00 | 3.8549 | 1.05441 |
| S6 | 317 | 1.00 | 5.00 | 4.2650 | 0.92391 |
| S7 | 317 | 1.00 | 5.00 | 3.9811 | 1.00613 |
| S8 | 317 | 1.00 | 5.00 | 4.2902 | 0.88106 |
| S9 | 317 | 1.00 | 5.00 | 4.0410 | 0.96204 |
| S10 | 317 | 1.00 | 5.00 | 3.8801 | 1.03643 |
| S11 | 317 | 1.00 | 5.00 | 4.0000 | 0.97760 |
| S12 | 317 | 1.00 | 5.00 | 3.8170 | 1.01786 |
| S13 | 317 | 1.00 | 5.00 | 4.2082 | 1.05281 |
| S14 | 317 | 1.00 | 5.00 | 3.8801 | 1.06356 |
| S15 | 317 | 1.00 | 5.00 | 3.8549 | 1.01465 |
| S16 | 317 | 1.00 | 5.00 | 3.9653 | 1.08005 |
| S17 | 317 | 1.00 | 5.00 | 4.1230 | 1.01601 |
| S18 | 317 | 1.00 | 5.00 | 4.1199 | 1.06356 |
| Valid N (listwise) | 317 |  |  |  |  |

Source: authors' compilation.

## Appendix B

**Table A2.** Age.

|  |  | Frequency | Percent | Valid Percent | Cumulative Percent |
|---|---|---|---|---|---|
|  | 21–29 | 22 | 6.9 | 6.9 | 6.9 |
|  | 30–39 | 126 | 39.7 | 39.7 | 46.7 |
| Valid | 40–49 | 67 | 21.1 | 21.1 | 67.8 |
|  | 50–59 | 102 | 32.2 | 32.2 | 100.0 |
|  | Total | 317 | 100.0 | 100.0 |  |

**Table A3.** Gender.

|  |  | Frequency | Percent | Valid Percent | Cumulative Percent |
|---|---|---|---|---|---|
|  | Male | 217 | 68.5 | 68.5 | 68.5 |
| Valid | Female | 100 | 31.5 | 31.5 | 100.0 |
|  | Total | 317 | 100.0 | 100.0 |  |

**Table A4.** Locality.

|       |         | Frequency | Percent | Valid Percent | Cumulative Percent |
|-------|---------|-----------|---------|---------------|--------------------|
| Valid | North   | 99        | 31.2    | 31.2          | 31.2               |
|       | Central | 174       | 54.9    | 54.9          | 86.1               |
|       | South   | 44        | 13.9    | 13.9          | 100.0              |
|       | Total   | 317       | 100.0   | 100.0         |                    |

**Table A5.** Provider–user.

|       |                                | Frequency | Percent | Valid Percent | Cumulative Percent |
|-------|--------------------------------|-----------|---------|---------------|--------------------|
| Valid | User of Financial Institutions | 214       | 67.5    | 67.5          | 67.5               |
|       | Financial Services Provider    | 103       | 32.5    | 32.5          | 100.0              |
|       | Total                          | 317       | 100.0   | 100.0         |                    |

**Table A6.** Level of expertise.

|       |           | Frequency | Percent | Valid Percent | Cumulative Percent |
|-------|-----------|-----------|---------|---------------|--------------------|
| Valid | Good      | 189       | 59.6    | 59.6          | 59.6               |
|       | Excellent | 128       | 40.4    | 40.4          | 100.0              |
|       | Total     | 317       | 100.0   | 100.0         |                    |

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
