# Peer review of "The Implications of Derisking: The Case of Malta, a Small EU State"

_jrfm, doi:10.3390/jrfm13090216_

Round 1

Reviewer 1 Report

 This is an interesting and well written study of de-risking in Malta, involving interviews with banking participants and the analysis of the data obtained using factor analysis and regressions. i have the following comments:

  • In the introduction the specific contributions of this study to the literature should be emphasised, including the empirical aspects.
  • Why would the increased costs of transferring money appreciate the real exchange rate, as suggested in S3 on page 20?
  • The data obtained from the questionnaires could do with some more discussion, including a table of summary statistics.
  • In the regression analysis, have the authors used robust standard errors to remove any heteroskedasticity?
  • In the regression model, all the explanatory variables are not significant, is there an explanation for this?
  • In the regression, it would be interesting to add an age squared term to pick up any non-linear relationship. Also is it possible to include some dummy variables for the different types of financial institutions that the interviewees represent?

Author Response

Thank you for your comments and suggestions, which has helped to make our paper stronger and improve clarity on specific areas. We have tried to answer each comment and suggestion made below yours (in blue and in the attached document) for ease of reference. All changes and additions to the manuscript where track changed.

Reviewer 2 Report

Please pay attention to punctuation and spelling.  Moreover, comments to the article are attached. 

Author Response

(The authors gave the same response as above.)

Round 2

Reviewer 2 Report

No comments